# High Foam Phenotypic Diversity and Variability in Flocculant Gene Observed for Various Yeast Cell Surfaces Present as Industrial Contaminants

**Catarina M. de Figueiredo** [1], **Daniella H. Hock** [1], **Débora Trichez** [1], **Maria de Lourdes B. Magalhães** [1,2], **Mario L. Lopes** [3], **Henrique V. de Amorim** [3] **and Boris U. Stambuk** [1,*]

1   Center of Biological Sciences, Department of Biochemistry, Federal University of Santa Catarina, Florianópolis 88040-900, SC, Brazil; catfigueiredo@gmail.com (C.M.d.F.); daniellahock@hotmail.com (D.H.H.); debora_trichez@yahoo.com.br (D.T.); maria.magalhaes@udesc.br (M.d.L.B.M.)
2   Center of Agroveterinary Sciences, Biochemistry Laboratory, Department of Food and Animal Production, State University of Santa Catarina, Lages 88520-000, SC, Brazil
3   Fermentec, Av. Antonia Pazinatto Sturion 1155, Piracicaba 13420-640, SP, Brazil; mario@fermentec.com.br (M.L.L.); amorim@fermentec.com.br (H.V.d.A.)
*   Correspondence: boris.stambuk@ufsc.br; Tel.: +55-48-3721-4449

**Abstract:** Many contaminant yeast strains that survive inside fuel ethanol industrial vats show detrimental cell surface phenotypes. These harmful effects may include filamentation, invasive growth, flocculation, biofilm formation, and excessive foam production. Previous studies have linked some of these phenotypes to the expression of *FLO* genes, and the presence of gene length polymorphisms causing the expansion of *FLO* gene size appears to result in stronger flocculation and biofilm formation phenotypes. We performed here a molecular analysis of *FLO1* and *FLO11* gene polymorphisms present in contaminant strains of *Saccharomyces cerevisiae* from Brazilian fuel ethanol distilleries showing vigorous foaming phenotypes during fermentation. The size variability of these genes was correlated with cellular hydrophobicity, flocculation, and highly foaming phenotypes in these yeast strains. Our results also showed that deleting the primary activator of *FLO* genes (the *FLO8* gene) from the genome of a contaminant and highly foaming industrial strain avoids complex foam formation, flocculation, invasive growth, and biofilm production by the engineered (*flo8Δ::Ble*^R/*flo8Δ::kanMX*) yeast strain. Thus, the characterization of highly foaming yeasts and the influence of *FLO8* in this phenotype open new perspectives for yeast strain engineering and optimization in the sugarcane fuel-ethanol industry.

**Keywords:** foam; flocculation; *FLO* genes; *Saccharomyces*; fuel-ethanol; *FLO8*

## 1. Introduction

The renewable Brazilian sugarcane first-generation fuel-ethanol industry is highly competitive, when compared with ethanol production processes from other crops (e.g., corn and sugarbeet), as it shows the highest percentage of greenhouse gas emissions' reduction, highest energy balance and yields per hectare, and the lowest production costs [1,2]. The productivity of the Brazilian fuel ethanol industry has increased steadily since 1975, an increase possibly due to several improvements, including the selection of new sugarcane varieties with an increased amount of sugarcane biomass per hectare and also the amount of ethanol produced from each ton of sugarcane, as the industrial sector responsible for this, including the fermentation process, has also reached high industrial efficiency [3–5].

In Brazil, over 70% of the milling and distilling companies perform the Melle–Boinot process, a fed-batch fermentation with high cell density that allows higher production stability when compared to the continuous mode [6,7]. At the end of each fermentation, the yeast cells are centrifuged, treated with diluted sulphuric acid, and over 90% of the yeast cells are reused from one fermentation to the next to ensure the high cell density

that contributes to very short fermentation times and, thus, the high productivity of the process [3,5,8]. However, as many industrial processes, the substrate (sugarcane juice and/or diluted molasses) is not sterile and, thus, a continuous input of contaminant microorganisms is observed [9,10]. While lactic acid bacteria are contaminants of concern in industrial ethanol fermentations [11,12], wild yeasts also affect the productivity of these fermentations, leading to decreased efficiency and stuck fermentations that cause plants to shut down for cleaning before beginning a new fermentation [13–16].

*Saccharomyces cerevisiae* is the predominant microorganism responsible for the efficient production of fuel ethanol [17], and studies of the microbiological dynamics of the industrial fermenters revealed a very rapid succession of yeast strains during fuel ethanol fermentations and, consequently, the original starter yeast (usually a commercial baker's yeast strain) is completely replaced by other strains in a matter of weeks [18–21]. Nevertheless, some yeast strains tend to dominate the fermenters, allowing the selection of suitable industrial strains with high fermentative capacity and viability to withstand the stressful industrial conditions, which includes high temperatures and osmotic pressures, high ethanol concentrations, low nutrients and pH, and process interruptions, to mention some stresses [4,19,22,23]. Another highly desired characteristic for the yeast cells used in this industrial process is that cells need to be in an unicellular planktonic state to allow efficient centrifugation for cell recycling, avoiding other operational problems (e.g., clogging, stuck/sluggish fermentations) that are a consequence of structured multicellular lifestyles (biofilms, premature/excessive flocculation) that this microorganism may adopt [19,24,25].

Another cell-surface phenotype, intensive foam formation, is also of great concern as highly foaming yeast strains do not allow the use of the fermenter total volume capacity, decreasing productivity and increasing costs due to the use of antifoam agents [3,19,26,27]. Moreover, it has been recently shown that industrial antifoam agents impair ethanol fermentation and decrease yeast cell viability, inducing a clear stress response in an industrial strain under conditions that simulate the Brazilian sugarcane-based ethanol production process with cell recycling [28]. While foam formation and stability are important in some fermented beverages like sparkling wines and beer [29,30], excessive foam production during fuel-ethanol fermentation is undesirable. Foam accumulates at the air-liquid interface and is the consequence of a complex process where $CO_2$ bubbles absorb a series of compounds (cells, proteins, carbohydrates, polyphenols, ions, and surfactants), which influence foam production and maintenance. Yeast glycoproteins present in the cell wall may be released during fermentation and, while their hydrophobic proteinaceous domains can interact with air bubbles, their hydrophilic mannose polysaccharides are exposed to the aqueous medium, contributing to foam formation and stabilization [30–32]. Approaches to stabilize beer foam include decreasing the secretion of proteinase A by the yeast cells, a protease known to hydrolyze proteins important for foam stability [33,34]. Yeast cells with a highly hydrophobic cell wall can also attach to $CO_2$ bubbles, contributing to the production of stable and rigid foams [35,36].

All these different phenotypic lifestyles that *S. cerevisiae* cells adopt are the consequence of the expression of specific cell-wall proteins that mediate cell–cell as well as cell–environment interactions [37,38]. The most important yeast proteins involved in yeast flocculation, biofilm formation, invasive growth, and other cell surface phenotypes are the adhesins or flocculins encoded by the *FLO* gene family. The genome of *S. cerevisiae* contains several telomeric *FLO* genes encoding for flocculins (e.g., *FLO1*, *FLO5*, *FLO9*, and *FLO10*), the *FLO11* gene (also called *MUC1*) located in the sub-telomeric region of chromosome IX, and the transcriptional activator *FLO8* located in the right arm of chromosome V [37–39]. Several studies have identified the *FLO1* and *FLO11* gene products as major participants in yeast cell-surface processes [40]. These adhesins belong to a large family of fungal cell-wall glycoproteins that usually share only a low degree of primary amino acid sequence similarity, but have a common overall architecture consisting of at least three different domains. The N-terminal domain that follows the secretion signal peptide is exposed at the cell surface and confers the recognition and binding of ligand molecules, a

middle domain that contains many tandem repeats extremely rich in serine and threonine (and also proline) residues that is highly glycosylated, and a C-terminal domain were the glycosylphosphatidylinositol (GPI) anchor is added for localizing the adhesins to the cell surface, although the adhesins may also be covalently linked to the non-reducing end of β-1,6-glucans of the cell wall via a GPI remnant [38,41,42].

The N-terminal domain differs significantly between *FLO11* and the other flocculin genes. The structure of the N-terminal domain of the flocculin encoded by *FLO11* (Pfam entry PF10182, https://pfam.xfam.org, accessed on 20 May 2020) is composed of three subdomains, including a hydrophobic apical region, a β–sandwich of the fibronectin type III domain, and the neck subdomain [43]. Two aromatic bands (containing tryptophan and tyrosine residues) that are present at the ends of the adhesion domain mediate homotypic and hydrophobic adhesion to a range of solid surfaces and the air–liquid interface of flor (or velum) in sherry-like wine fermentations [42–46]. The N-terminal domain of the other flocculins (e.g., *FLO1* or *FLO5*) have a lectin-like structure (PA14 domain, Pfam entry PF07691) that binds (in a $Ca^{2+}$-dependent manner) to mannose oligosaccharides present in the yeast cell surface [42,47,48]. Consequently, *FLO11* and the other *FLO* genes play distinct roles in adhesion, flocculation, invasive growth, biofilm formation, and other cell surface phenotypes of yeast cells [40,49]. A further level of complexity is determined by the serine/threonine-rich middle domain, which contains intragenic tandem repeats of DNA sequences that trigger recombination, causing variations in gene size, and the expansion of this middle domain contributes to stronger flocculation, biofilm formation, and other cell surface phenotypes of yeast cells [39,50,51].

The first yeast gene described as specifically involved in foam formation was the *AWA1* gene, encoding for a cell-wall GPI-linked and highly glycosylated serine/threonine-rich protein from sake-producing yeasts [52]. As with *FLO* genes, the serine/threonine-rich domain in *AWA1* can also vary in size, contributing to different foaming phenotypes of sake yeasts [53,54]. More recently, primers based in the *AWA1* gene allowed the identification of two other highly homologous genes, the *FPG1* gene responsible for foam formation in a wine yeast strain [55] and the *CFG1* gene involved in foam production by *S. pastorianus* lager brewing yeasts [56].

In the present report, we analyzed several cell-surface properties (flocculation and cell-wall hydrophobicity, foam and biofilm production, invasive growth) of a group of 15 wild and indigenous yeast strains isolated from sugarcane industrial fuel-ethanol production plants, including strains with good or bad (e.g., premature flocculation, excessive foam production) industrial characteristics. We also determined gene length polymorphisms in some adhesin genes (*FLO1* and *FLO11*) and analyzed possible correlations with the phenotypic diversity found in this group of contaminant industrial yeast strains. Finally, we showed that a highly foaming industrial fuel-ethanol yeast strain loses its capacity to produce foam when this *FLO8* gene is deleted, indicating that the expression of flocculins is involved in problematic (abundant and persistent) foam formation during industrial fermentation by this contaminant *S. cerevisiae* yeast.

## 2. Materials and Methods

### 2.1. Strains, Media, and Growth Conditions

The *S. cerevisiae* industrial strains studied were isolated during fuel ethanol production in Brazilian distilleries [19] and included two reference strains (CAT-1 and PE-2) widely used by many distilleries [23,57,58], as well as 13 strains (BAT-1, FT278, FT279, FT281, FT517, FT540, FT645, FT697, FT699, FT705, FT714, FT718, FT859) with reported premature flocculation/sedimentation and/or intensive foam production during industrial fuel-ethanol production [19]. The laboratory strains S288C (*MATα mal gal2 mel flo1 flo8-1 hap1 SUC2*) and CEN.PK2-1C (*MATa MAL2-8$^c$ ura3-52 his3Δ1 leu2-3_112 trp1-289*) were also used for mutant construction and amplification of probes for Southern analysis. Yeasts were cultivated on YPD medium (2% peptone, 1% yeast extract, and 2% glucose) or molasses-based medium containing 20 g/L total sugars from sugarcane molasses, 0.5%

yeast extract, 0.5% $(NH_4)_2SO_4$, and 0.2% $KH_2PO_4$. When required, 0.3 or 2% agar and 200 mg/L geneticin (G-418) sulfate (Invitrogen, Thermo Fisher Scientific Inc., Sinapse Biotecnologia, São Paulo, SP, Brazil) or 100 mg/L zeocin (Invivogen, San Diego, CA, USA) were added to the medium. Cells were grown aerobically at 28 °C with shaking (160 rpm) in cotton-plugged Erlenmeyer flasks filled to 1/5 of the volume with medium.

### 2.2. Foam Quantification

Yeast cells were cultivated in rich YPD medium until $1 \pm 0.5$ g (dry cell weight, DCW)/L, harvested by centrifugation ($7.000 \times g$, 5 min) and washed twice with cold, sterile water. A cellular suspension of 10 g DCW/L was prepared in 5 g/L $Na_2SO_4$ and added to molasses-based medium (final sugar and other components' concentrations as indicated above), incubated at room temperature under constant agitation (100 rpm) during 8 h, and foam formation was volumetrically monitored during the fermentation. Foam formation was expressed as the percentage of foam volume in relation to the fermentation media volume.

### 2.3. Flocculation Tests

To test cell–cell adhesion, flocculation assays were performed based on the protocol described by Wang et al. [59]. Cells were grown in molasses medium until the late exponential growth phase ($3 \pm 0.5$ g DCW/L), harvested by centrifugation, and washed twice with 50 mM sodium citrate, 50 mM EDTA, and pH 3.0 buffer, followed by two washings with cold, distilled water. After centrifugation, cells were suspended (5 g DCW/L) in 100 mM sodium acetate, pH 4.0 buffer, and their ability to flocculate in the presence or absence of 10 mM $CaCl_2$ was determined. Briefly, cells were vortexed during 10 s and allowed to stand for 10 min. Aliquots of 30 μL taken from below the meniscus (at 0 and 10 min) were used to determine (after proper dilution) the cell density at 570 nm. Flocculation was expressed as the percentage of sedimented cells after 10 min, relative to the initial value.

### 2.4. Cellular Hydrophobicity Assay

Cells were grown in molasses-based medium as described above, washed twice in cold, distilled water, and suspended in water to an absorbance of 0.8 at 570 nm. Hexane was added to the cellular suspensions at a proportion of 1:3 (v:v) and vigorously mixed. The relative difference in cell density of the aqueous phase was determined by measuring the absorbance at 570 nm before and after vortexing and considered as the hydrophobicity percentage of the cell surface.

### 2.5. Biofilm and Invasive Growth Determination

Biofilm formation was analyzed as described by Reynolds and Fink [60]. The yeast strains were inoculated onto YPD plus 3 g/L agar plates with a sterile toothpick. The plates were maintained resting at room temperature, and the yeast colony was measured and photographed after 5 days. For invasive growth assays, the cells were streaked onto standard YPD plus 2% agar plates, grown at 28 °C for 5 days, and then photographed. Tap water was used to gently wash non-invasive cells from the surface of the agar, and then the plate was photographed again [40].

### 2.6. Amplified Fragment Length Polymorphism Analysis

Standard methods for DNA manipulation and analysis were employed [61]. Approximately 50 ng of DNA from the different industrial yeast strains, purified using YeaStar columns (Zymo Research, Irvine, CA, USA), according to the manufacturer's recommendations, and 20 pmol of the primers (Table 1) for the *AWA1* (AWA1F and AWA1R), *FLO1* (FLO1F and FLO1R), and *FLO11* (FLO11F and FLO11R) genes were used in 25 μL PCR reaction mixture containing standard PCR High Fidelity buffer, 10 mM dNTPs, 1.5 mM $MgCl_2$, and 1 U Phusion High-Fidelity DNA polymerase (Finnzymes, Thermo Fisher Scientific Inc., Sinapse Biotecnologia, São Paulo, SP, Brazil). The following PCR program was used

to amplify fragments: denaturation (10 s at 98 °C); primer annealing (30 s at 60 °C) and primer extension (3 min at 72 °C), for 30 cycles. PCR products were fractionated on 0.85% agarose gels in 40 mM Tris-acetate buffer (pH 8.5) plus 2 mM EDTA and visualized with 0.5 µg/mL ethidium bromide staining. Sizes of the amplified fragments were calculated using 1 Kb DNA ladder (New England Biolabs, Uniscience, Osasco, SP, Brazil) as marker.

**Table 1.** Plasmids and primers used in this study.

| | Relevant Features or Sequence |
|---|---|
| Plasmids: | |
| pUG6 | *LoxP-KanMX6-LoxP* [62] |
| pUG66 | *LoxP-Ble*R*-LoxP* [62] |
| Primers: | |
| AWA1F | CCGAAGCACTTGCAAAGGATGG |
| AWA1R | GGGAGTTGGAAGCAGTTGCGC |
| FLO1F | GGCAGTCTTTACACTTCTGGCAC |
| FLO1R | AGTATTGGTAGTCGTTTCAGCAGC |
| FLO11F | GATGTGACTTCCGTTTCTTGGG |
| FLO11R | CCGTAGTGACTGATGGAGTTGGAG |
| HbFLOsF | CAGAAACAACAAAACAACCA |
| HbFLOsR | TTAAATTAATTGCCAGCAATAAG |
| HbFLO11F | CCGTCGTTACTACTGAGTATTC |
| HbFLO11R | TAGAATACAACTGGAAGAGCG |
| INFLO8F | GTTATAAAGTGAATAGTTCGTATC |
| INFLO8R | CGATCTCAATTTACTGGATAC |
| LIM1FLO8F | CACGGCACGTTACTAATTAG |
| LIM1FLO8R | CAGATTAAGGTTTATGATATTTG |
| LIM2FLO8F | GTGACCATTTTTTACTCCTG |
| LIM2FLO8R | CAAGCTACTTCAATGAGTGTAC |
| M1FLO8F1 | GCCCTGGGAGTGGGATACTGAGGAATTAAACGTTTATGCAGGCCCAGCTGAAGCTTCGTACGC |
| M1FLO8R1 | CCAATCATTCCGGCACATCCACCTTGATCAAATATCATAAACCGCATAGGCCACTAGTGGATC |
| M2FLO8F1 | CGCATTCCGTGGTAGAATGAGTTATAAAGTGAATAGTTCGCCAGCTGAAGCTTCGTACGC |
| M2FLO8R1 | CGTAACTCCATTCTCCTAGCTTTTTATTATGTTTCCTGTCGCATAGGCCACTAGTGGATC |
| SDKANF | CCGCGATTAAATTCCAACAT |
| SDKANR | CGAGCATCAAATGAAACTG |
| SDBLEF | GAGATCTGTTTAGCTTGCCT |
| SDBLER | GTTAGTATCGAATCGACAGCA |
| V1FLO8F | GGCTCTAGTAGTAACAAAAATAG |
| V3FLO8F | CTGAGGGACCTATTGTATGC |
| VKanRR | GGAATCGAATGCAACCGG |
| VBLERF | CCTTCTATGAAAGGTTGGGC |

Underlined sequences allow amplification of the transformation modules present in plasmids pUG6 and pUG66 [62].

### 2.7. Construction of Flo8∆ Knockout Strains

Yeast transformation was performed by the lithium acetate method [44]. The *FLO8* gene was deleted according to the polymerase chain reaction (PCR)-based gene replacement procedure [62]. Briefly, the *kanMX* cassette from plasmid pUG6 (Table 1) was amplified with primers M1FLOF1 and M1FLO8R1 (Table 1), and the resulting PCR product of 1779-bp was used to transform competent yeast cells [63] from the laboratory strain S288C. After 2-h cultivation on YP-2% glucose, the transformed cells were plated on the same medium containing G-418 and incubated at 28 °C. G-418-resistant isolates were tested for proper genomic integration of the *kanMX* cassette at the *FLO8* locus by analytical colony PCR using primers (V1FLO8F and INFLO8R; Table 1) that yielded a 2857-bp fragment from a normal *FLO8* locus or with primers (V3FLO8F and VKanRR, Table 1) that amplified a 1394-bp fragment if the *kanMX* cassette was correctly integrated at this locus and replaced the *FLO8* gene, yielding strain S288C-*flo8∆::kanMX*. Using the same strategy above, the *FLO8* gene was deleted from the genome of the laboratory strain CEN.PK2-1C using primers M2FLO8F1 and M2FLO8R1 (Table 1) to amplify the *Ble*R gene from plasmid pUG66 [62].

The resulting PCR product of 1344-bp was used to transform competent yeast cells and zeocin-resistant isolates were tested for proper genomic integration of the *Ble*[R] gene at the *FLO8* locus by Southern analysis (see below) or analytical colony PCR using primers V1FLO8F and INFLO8R, as described above, or with primers LIM2FLO8R and VBLE[R]F (Table 1) that amplified a 956-bp fragment if the *FLO8* gene was deleted and replaced by *Ble*[R], yielding strain CEN.PK2-1C-*flo8Δ::Ble*[R]. To delete both copies of the *FLO8* gene present in the industrial diploid strain FT859, we used a long-flanking-homology approach [62] by using primers LIM1FLO8F and LIMFLO8R (Table 1) to amplify a 2118-bp fragment form the DNA of strain S288C-*flo8Δ::kanMX* or primers LIM2FLOF and LIM2FLO8R (Table 1) to amplify a 1646-bp fragment from the DNA of strain CEN.PK2-1C-*flo8Δ::Ble*[R]. After selection of transformants in G-418 and zeocin plates, the proper genomic integration of the *kanMX* and *Ble*[R] genes at the *FLO8* locus was confirmed by Southern analysis (see below) or analytical colony PCR, as described above, yielding the industrial-modified strain CFY02B-B (FT859 *flo8Δ::kanMX/flo8Δ::Ble*[R]).

### 2.8. PFGE, Chromosome Blotting, and Hybridization

Yeast chromosomes were prepared as previously described [64] from 1 mL of yeast cells pre-grown in YPD medium and collected at the stationary phase of growth. Cells were washed with 10 mM Tris-HCl, pH 7.5, containing 50 mM EDTA, and resuspended in the 0.4 mL of 4 mM Tris-HCl, pH 7.5, containing 95 mM EDTA, 130 μg/mL of Zymolyase 20 T (Sigma-Aldrich Brasil Ltda., São Paulo, SP, Brazil), and 0.7% of molten (42 °C) low-melting-point agarose. After solidification in a mold (Amersham Pharmacia Biotech do Brasil Ltda., São Paulo, SP, Brazil), the agarose blocks were immersed in 10 mM Tris-HCl, pH 7.5, containing 0.5 M EDTA, and incubated at 37 °C for 8 h. Following a subsequent incubation in 10 mM Tris-HCl, pH 9.5, containing 0.5 M EDTA, 1% *N*-lauroylsarcosine, and 2 mg/mL proteinase K at 50 °C overnight, the blocks were washed in 10 mM Tris-HCl, pH 7.5, containing 50 mM EDTA, and stored at 4 °C in the same buffer. Each low-melting-point agarose block was transferred to a 1% agarose gel in 50 mM Tris-HCl, pH 8.3, containing 50 mM boric acid and 1 mM EDTA. Pulsed field gel electrophoresis (PFGE) was performed at 8 °C using a Gene Navigator pulsed field system (Amersham Pharmacia Biotech Pharmacia Biotech do Brasil Ltda., São Paulo, SP, Brazil) for a total of 27 h at 200 V. The pulse time was stepped from 70 s after 15 h to 120 s for 12 h. Following electrophoresis, the gel was stained with ethidium bromide and photographed (Gel Doc™ XR, Bio-Rad Laboratories, Hercules, CA, USA).

The chromosomes separated by PFGE were transferred to a nylon filter (Hybond-N[+], GE Healthcare, Barueri, SP, Brazil) by capillary blotting [61], and labeling of DNA probes (see below), including the pre-hybridization, hybridization, stringency washes, and chemiluminescent signal generation and detection was performed by using a AlkPhos kit (GE Healthcare, Barueri, SP, Brazil) as recommended by the manufacturer. After hybridization, an autoradiography film (Hiperfilm™ ECL, Kodak, GE Healthcare, Barueri, SP, Brazil) was exposed to the membrane for 1 to 2 h before it was developed. Images were obtained with Image Lab Software (Gel Doc™ XR, Bio-Rad Laboratories, Hercules, CA, USA) and annotated with Microsoft PowerPoint. Probes were generated by PCR using DNA from strain S288C or plasmids pUG6 and pUG66 as templates and primers (Table 1) INFLO8F and INFLO8F for the *FLO8* gene, HbFLOsF and HbFLOsR for *FLO1*, HbFLO11F and HbFLO11F for the *FLO11* gene, SDKanF and SDKanR for *kanMX*, and SDBleF and SDBleR for the *Ble*[R] gene, respectively.

### 3. Results

### 3.1. Foam Production by Industrial Yeast Strains

We analyzed 15 industrial yeast strains isolated during fermentative fuel-ethanol production processes in Brazil. This group included two strains (PE-2 and CAT-1) with good industrial fermentation characteristics, such as non-flocculent cells with high fermentative capacity, dominance of the industrial fermenters, and production of very small amounts

of foam during industrial fuel-ethanol production, and, consequently, these two strains are used by more than half of the industrial fuel-ethanol companies in Brazil [4,19,23]. The other 13 industrial strains (BAT-1 and FT278 through FT859) showed some undesired industrial characteristics, such as premature flocculation and/or sedimentation or excessive foam production during industrial fermentation processes, generally leading to problematic/stuck fermentations with residual sugars not completely fermented in the vats.

Figure 1 shows the kinetics of foam production during fermentation of molasses-based medium by three representative strains (CAT-1, BAT-1, and FT281). Strain CAT-1 produced less than 10% (*v/v*) of foam during fermentation, while strains BAT-1 and FT281 produced significant amounts of foam (more than 60% for strain BAT-1 and up to 90% for strain FT281), with significant differences in the kinetics of foam production. Strain FT281 started to produce foam early during fermentation, and the maximum production peaked after 2 h, but, soon after that, the foam started to decrease and disappeared after 4 to 5 h of fermentation (Figure 1). Strain BAT-1 started to produce foam earlier during fermentation, but, once the foam accumulated, it persisted all along the 8 h of fermentation (Figure 1). We observed the presence of high amounts of yeast cells attached to this highly stable, thick, dense, and persistent foam (data not shown). Figure 2 shows the maximum amount of foam produced by all the 15 industrial yeast strains analyzed. Some of them (strains CAT-1, PE-2, FT279, FT540, and FT718) were considered as poor foam producers (less than 20% of the total volume), while the other 10 strains produced 35% or more of foam during fermentation, the majority with a stable and persistent foam all along the fermentation (the exceptions were strains FT281, FT705, and FT714, see Figures 1 and 2).

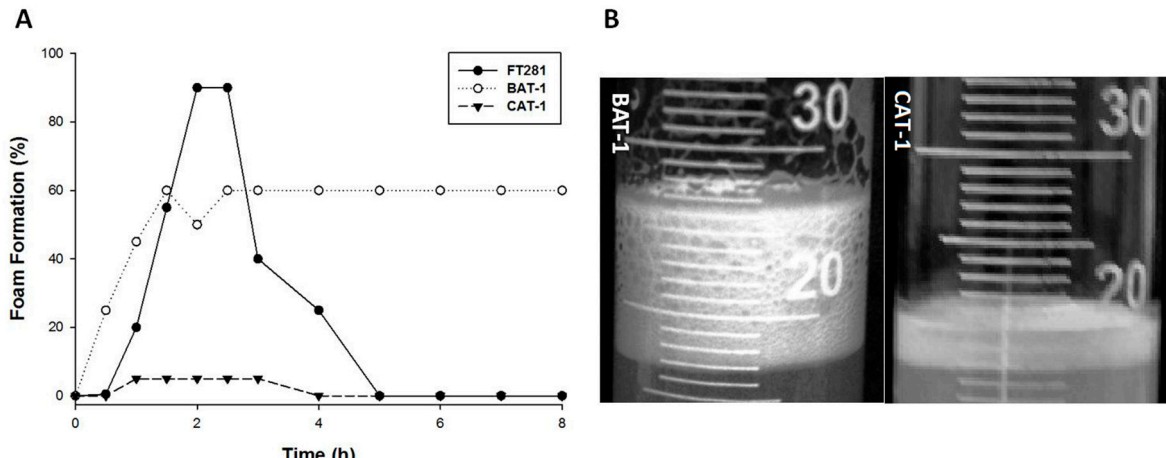

**Figure 1.** Foam production during fermentation. (**A**) Kinetics of foam formation (volume of foam as a percentage of the media volume) during fermentation of molasses-based medium containing 20 g/L of total sugars by three representative industrial strains. (**B**) Foam produced after 3 h of fermentation by strain BAT-1 (left panel) or strain CAT-1 (right panel).

### 3.2. Flocculation and Cellular Hydrophobicity of Industrial Yeast Strains

Yeast strains were grown in molasses-based medium and flocculation assays were performed in the presence or absence of 20 mM $Ca^{2+}$, and at least three different patterns of yeast flocculation were observed (Figure 3). Six industrial strains showed a high cellular sedimentation (>60% of flocculation), which was mostly calcium independent and only one of them (strain FT718) was not a strong foam producer (see Figure 2). A second group included four industrial yeast strains that presented calcium-dependent flocculation, as the sedimentation rate in absence of calcium was lower than 30%, but increasing to approximately 50% after $Ca^{2+}$ addition (Figure 3). Finally, the remaining five strains (including strains CAT-1 and PE-2) were either non-flocculent or showed an insignificant flocculation rate (less than 30%) in the presence or absence of calcium.

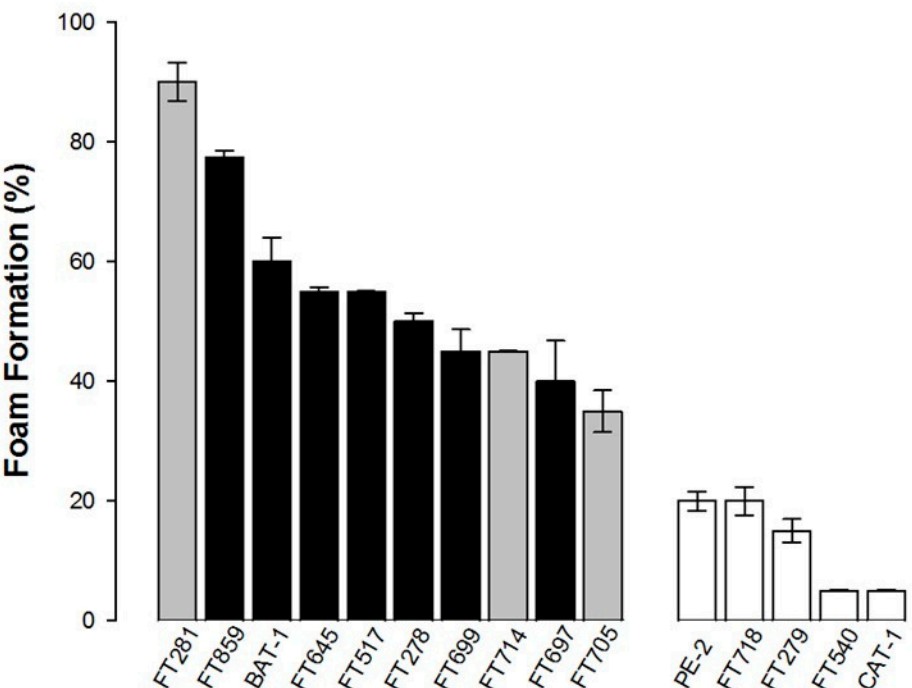

**Figure 2.** Foam produced during fermentation (volume of foam as a percentage of the media volume) by the analyzed industrial fuel-ethanol yeast strains. White columns are strains that produce small amounts of foam, gray columns are strains that produce a non-persistent foam, and persistent foaming strains are in black columns.

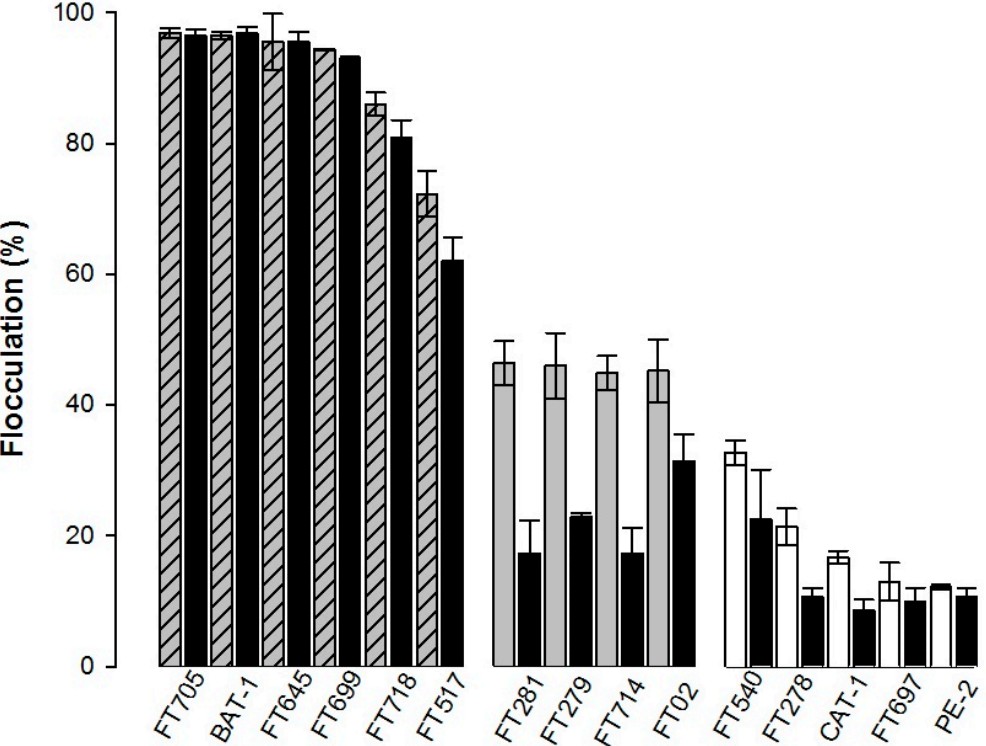

**Figure 3.** Cellular flocculation by the analyzed industrial fuel-ethanol yeast strains. Flocculation was determined as the percentage of sedimented cells after 10 min in rest, in the absence (black bars) or presence (white, grey, and striped grey bars) of 20 mM $Ca^{2+}$.

In order to quantify the cell wall hydrophobicity of the industrial strains, we analyzed the cellular partitioning into an organic solvent such as hexane. Our results showed a significant variability in cellular hydrophobicity among the industrial yeast strains (Figure 4), but, again, strains CAT-1 and PE-2 were the ones with the lowest cellular hydrophobicity. Among those strains with high cellular hydrophobicity (60% or more of cells partition into the organic solvent), it is worth to note that all of them had strong flocculation phenotypes (see Figure 3) and only one (strain FT718) was a poor foam producer (see Figure 2).

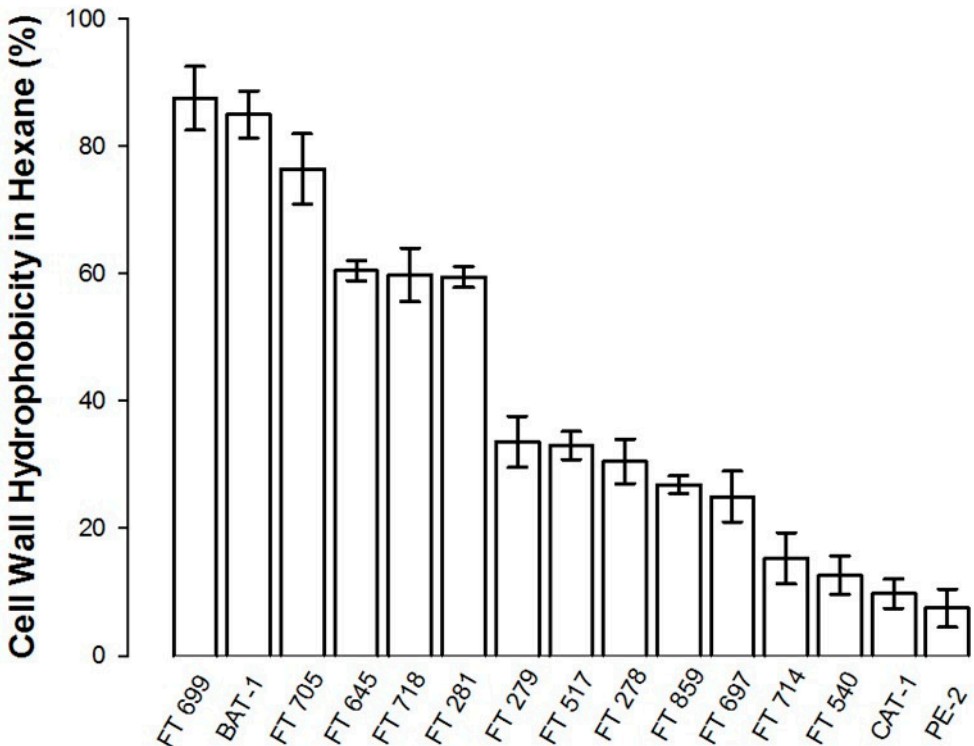

**Figure 4.** Cellular hydrophobicity of the analyzed industrial fuel-ethanol yeast strains. The hydrophobicity of the cells suspended in water was quantified by means of a biphasic partitioning assay using hexane as hydrophobic solvent.

### 3.3. Invasive Growth and Biofilm Formation by Industrial Yeast Strains

Figure 5 shows some patterns of invasive growth and biofilm formation by the industrial strains and in Table 2 are scored the size of mats and the degree of invasiveness into YPD agar plates for all strains. The industrial strains CAT-1 and PE-2 showed significant attachment to the agar surface and, while the other strains were either similar or even less invasive than these two strains, only one (strain FT278) attached firmly into the plate surface. Despite the highly variable size of the biofilms formed by the strains (diameters ranging from 1.2 to 4.5 cm after 5 days), the mats they produced were smooth and did not form highly structured "fluffy" biofilms, as described earlier for several other laboratory and wild yeast strains [60,65,66]. Nevertheless, biofilm formation seems to be directly correlated to flocculation, since all strong flocculent industrial strains were able to form biofilms larger than 3.5 cm and all non-flocculant strains formed biofilms smaller than 1.5 cm.

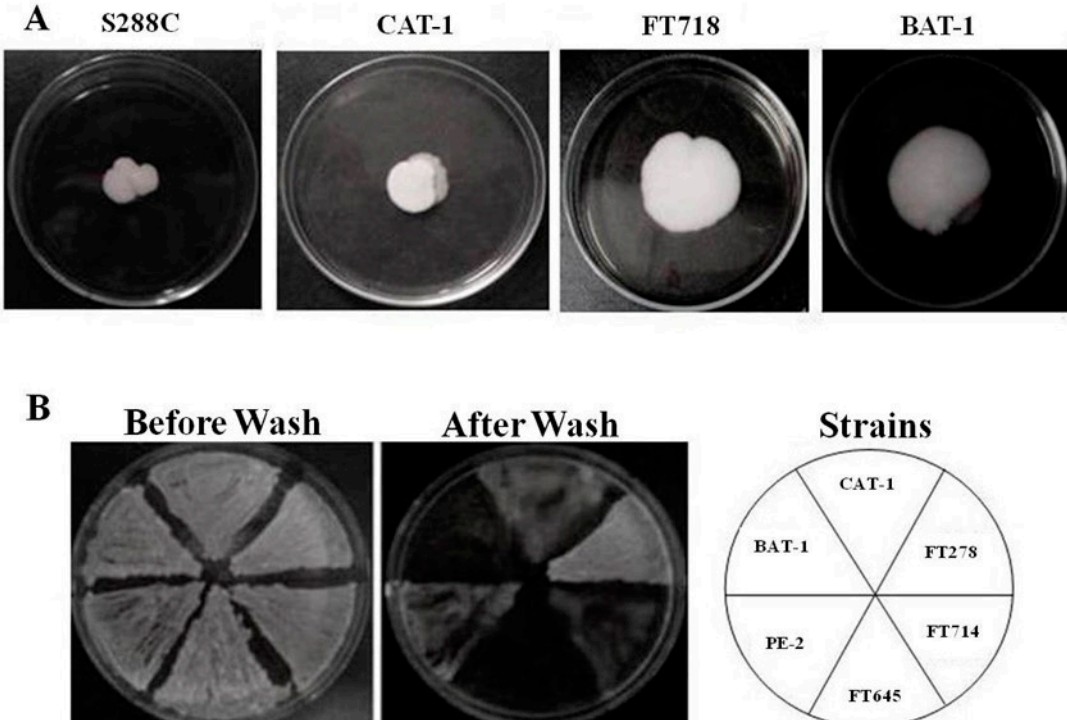

**Figure 5.** Biofilm formation and invasive growth by the analyzed industrial fuel-ethanol yeast strains. (**A**) Examples of biofilms (mats) produced by the indicated strains after 5 days in low agar (0.3%) plates. (**B**) Invasive growth, scored as adhesion to agar surfaces, was determined by patching the yeast cells on YPD plates and incubation for 5 days. The plate was documented before and after gentle washing with water. Strains BAT-1 and FT645 were considered non-adhesive ("−" in Table 2), while the other strains showed different strengths of invasive growth (scored "+" to "+++" in Table 2).

**Table 2.** Biofilm, invasive growth, and *FLO* gene polymorphisms of the industrial strains analyzed in this study.

| Industrial Yeast Strains: | Biofilm (cm) [1] | Invasive Growth [2] | PCR Product [3] (Kb) for Gene: | |
|---|---|---|---|---|
| | | | *FLO1* | *FLO11* |
| CAT-1 | 2.0 | ++ | 1.3 | 2.1 |
| PE-2 | 2.0 | ++ | 0.8 | 0.8 |
| BAT-1 | 4.5 | − | 3.2 | 2.7 |
| FT278 | 1.5 | +++ | 2.9 | 2.4 |
| FT279 | 3.0 | − | 2.9 | 2.5 |
| FT281 | 1.5 | − | 5.0 | 2.8 |
| FT517 | 3.5 | ++ | − | 2.6 |
| FT540 | 1.2 | + | 3.2 | 0.5 |
| FT645 | 4.0 | − | 3.4 | 2.9 |
| FT697 | 1.5 | ++ | 4.9 | − |
| FT699 | 4.0 | − | − | 2.7 |
| FT705 | 4.0 | + | − | 2.7 |
| FT714 | 3.0 | + | − | 2.6 |
| FT718 | 4.5 | + | − | 2.0 |
| FT859 | 1.6 | ++ | 3.0 | 2.6 |

[1] Diameter of the mat/biofilm. [2] "−" indicates absence of a characteristic (or gene), and "+" to "+++" the presence and strength of the characteristic (see Figure 5). [3] The expected amplified fragment, based on the genome of strain S288C, is 4.3 Kb for *FLO1* and 3.1 Kb for *FLO11*.

### 3.4. Presence and Flocculin Gene Size Polymorphisms

We designed primers to amplify the highly variable and repetitive serine/threonine-rich middle domain present in several GPI-linked cell-wall proteins (Table 1). Considering the highly foaming phenotypes present by some members of this group of industrial

yeast strains, we initially tried to amplify and verify fragment length polymorphisms in the already described *AWA1* foaming gene [52–54]. However, we could not amplify any fragment for this gene from the DNA of any of the industrial yeast strains analyzed. Thus, we focused or analysis on two other important flocculin genes, *FLO1* and *FLO11* (Table 2). The *FLO1* gene was found in 66% of the industrial yeast strains analyzed and presented significant length polymorphism in the serine/threonine-rich middle domain among the strains (sizes varied from 0.8 to 5.0 Kb), and only two strains (FT281 and FT697) showed PCR fragments longer that the expected 4.3 Kb fragment that is predicted from the known genome of strain S288C. Regarding *FLO11*, only strain FT697 showed no amplified fragment, and the other analyzed industrial strains showed DNA fragments of varying length polymorphisms (from 0.5 to 2.9 Kb), all of them with lower sizes than the expected size of 3.1 Kb obtained for this gene from the genome of strain S288C.

Nevertheless, we analyzed possible significant correlations (using STATISTIX 8.0® Analytical Software, Tallahassee, FL, USA) between the size polymorphisms of these two genes and the phenotypic variability shown by the corresponding industrial strains (Figures 1–5, Table 2). There was a significant correlation of *FLO11* length polymorphisms and foam production, cellular hydrophobicity, and flocculation ($r^2 \geq 0.50$ and $p \leq 0.05$, Spearman nonparametric test). For *FLO1* length polymorphisms, the more significant correlation observed was with cellular hydrophobicity ($r^2 \geq 0.62$ and $p \leq 0.05$) and, to a lesser extent, with foam production ($r^2 \geq 0.42$ and $p \leq 0.05$). Consequently, there was also a significant correlation between foam production during fermentation and the cellular hydrophobicity ($r^2 \geq 0.50$ and $p \leq 0.05$), while this last parameter was significantly correlated with flocculation (in the presence or absence of calcium, $r^2 \geq 0.70$ and $p \leq 0.05$) and biofilm formation ($r^2 \geq 0.42$ and $p \leq 0.05$, Pearson parametric test).

*3.5. Deletion of FLO8 Reduces Foam Production by an Industrial Yeast Strain*

Considering the correlation between *FLO11* (and also *FLO1*) gene length polymorphisms and foam production during fermentation by the industrial yeast strains, we decided to see the phenotypic consequences of deleting the *FLO8* gene, encoding one of the key transcriptional activators of the *FLO1* and *FLO11* genes [40,67], in one of the industrial yeast strains (FT859), a strain that produces large amounts of persistent foam (Figure 2), while in the other characteristics we analyzed it had intermediate (e.g., flocculation and invasive growth) to low (e.g., cellular hydrophobicity and biofilm formation) values (Figures 3–5, Table 2). We initially deleted the *FLO8* gene from two laboratory yeast strains (using different markers, see Materials and Methods) and afterward we used the long-flanking-homology approach [62] to delete this gene from the diploid genome of the industrial strain FT859. Figure 6 shows a PFGE and blotting analysis of the strains, confirming that the *FLO8* gene was specifically removed from the genome of the industrial-modified strain CFY02B-B (FT859 *flo8Δ::kanMX/flo8Δ::BleR*) and also showing the presence of both the *FLO1* (chromosome I) and *FLO11* (chromosome IX) genes in its genome.

Strain CFY02B-B (FT859 *flo8Δ/flo8Δ*) had no difference in cellular hydrophobicity when compared to strain FT859, but showed a pronounced drop in the flocculation capacity (15 and 20% of sedimented cells in the absence and presence of $Ca^{2+}$, compared to 32 and 45% obtained with strain FT859, see Figure 3), biofilm formation (mats of 0.9 cm of diameter, compared to 1.6 cm for strain FT859, see Table 2), invasive growth (strain CFY02B-B lost its capacity to adhere to the agar surface, while strain FT859 does, see Table 2) and, as shown in Figure 7, also lost its capacity to produce high amounts of foam during fermentation, indicating that induction of *FLO* genes by *FLO8* is required for foam production by this contaminant industrial yeast strain.

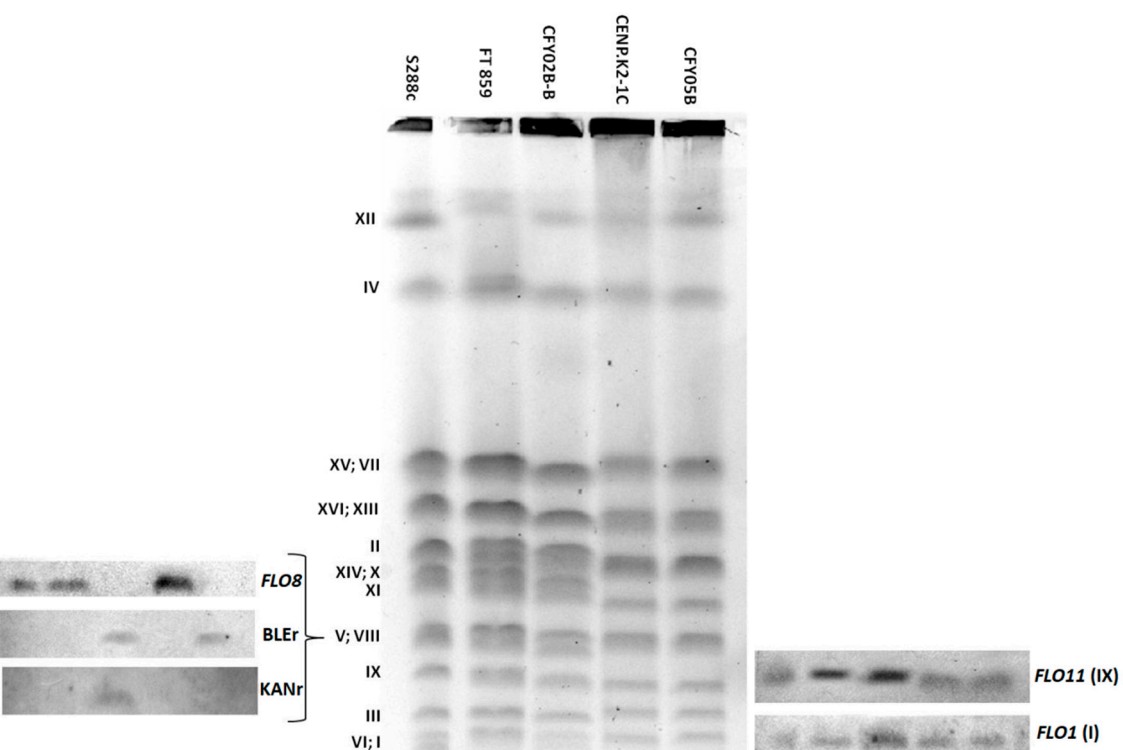

**Figure 6.** Confirmation of *FLO8* deletion in the industrial fuel-ethanol strain FT859. The middle panel shows the separation of chromosomes of the yeast strains by PFGE and staining with ethidium bromide, while the detection of *FLO1* and *FLO11* genes (right panels) or the presence/absence of the *FLO8*, *Ble*^R (BLEr), and *kanMX* (KANr) genes (left panels) in their genomes was determined after the chromosomes were blotted onto a nylon membrane and hybridized with specific probes. The Roman numerals are the chromosome numbers based on S288C chromosomes. In line 1 is strain S288C, in line 2 strain FT859, in line 3 strain CFY02B-B (strain FT859, but *flo8Δ::Ble*^R/*flo8Δ::kanMX*), in line 4 strain is CEN.PK2-1C, and in line 5 is strain CEN.PK2-1C -*flo8Δ::Ble*^R.

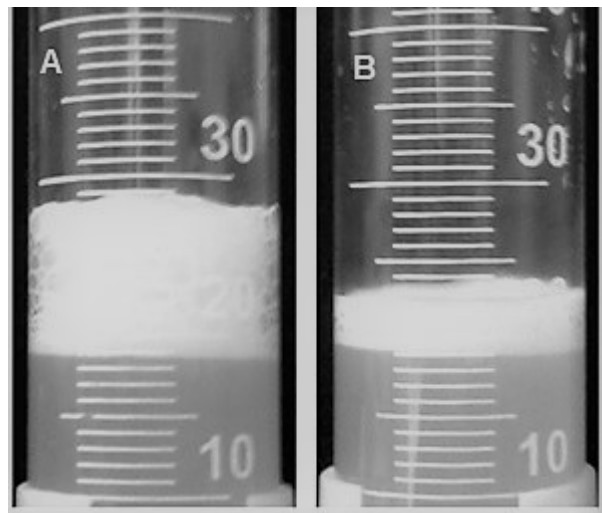

**Figure 7.** Deletion of *FLO8* prevents foam production by the industrial fuel-ethanol strain FT859. The foam produced by strain FT859 (**A**) or its isogenic *flo8Δ::Ble*^R/*flo8Δ::kanMX* strain CFY02B-B (**B**) was documented after 3 h of fermentation.

## 4. Discussion

The analysis of hundreds of indigenous *S. cerevisiae* strains isolated from industrial sugarcane fuel-ethanol producing plants in Brazil revealed that more than 67% of the strains

produced excessive amounts of foam. Only 20% of the strains did not present undesirable features, including foam, premature flocculation or inability to metabolize all sugar from the medium [19]. Yeasts growing in nature usually form organized multicellular structures (colonies, biofilms) with specific properties that do not normally arise under laboratory conditions. The cooperation of yeast cells and their growth within a community is beneficial in harmful or stressful conditions and might contribute to successful colonization of new environments. Consequently, these morphological transitions from a single-cell budding yeast that forms into organized structures are regulated by several sensing/signaling pathways that allow the formation of pseudohyphae and biofilms and promote their adhesion to other cells or surfaces [49,68,69].

In the case of the Brazilian sugarcane fuel-ethanol production plants there is not only several stressful conditions during the industrial process, but sucrose, the main sugar found in sugarcane juice that the cells will have to ferment [1], is one of the strongest inducers of pseudophyphal growth and filamentation in yeasts [70,71]. For example, after serial transfers of yeast cells in low sucrose-containing medium, of 12 evolved clones, 11 formed multicellular clumps through incomplete cell separation [72], due mainly to nonsense or frameshift mutations in the *ACE2* gene, encoding for the transcription factor required for septum destruction after cytokinesis [73–75]. Indeed, even a good industrial fuel-ethanol strain like PE-2 can eventually show recurrent appearance of mutants displaying a mother–daughter cell separation defect resulting in rough colonies in agar media and fast sedimentation in liquid culture [76]. This diploid strain was shown to be heterozygous for a frameshift mutation in the *ACE2* gene (*ACE2/ace2-A7*) and, thus, loss-of-heterozygosity events at the region of chromosome XII where *ACE2* is located can produce *ace2-A7/ace2-A7* cells with the undesirable phenotype of cellular aggregation [76]. Nevertheless, other reports indicate that mutations that increase the expression of the flocculin genes *FLO1* and *FLO11* are the favored routes to flocculation in yeasts [75,77].

Our results showed a great diversity of morphological and cell-surface-associated phenotypes in the group of 15 wild and indigenous yeast strains isolated from industrial fuel-ethanol production processes in Brazil that we analyzed. We not only found $Ca^{2+}$-dependent and -independent flocculation phenotypes, but also some strains showing significant invasive growth, a characteristic of haploid yeast cells not described in diploid strains (as are the industrial strains analyzed). While most laboratory strains do not show these dimorphic transitions or flocculate due to a mutation that inactivates the transcriptional activator encoded by the *FLO8* gene [39,40,78], the known genome of two Brazilian industrial strains included in our study (strains CAT-1 and PE-2, see [57,58]) revealed that these strains have a functional *FLO8* gene, but are non-flocculent and do not produce foam due to absence or significant polymorphisms in the coding regions of their flocculin *FLO* genes (including *FLO1* and *FLO11*, see Table 2).

Although we did not perform a comprehensive analysis of the *FLO* genes present in the industrial fuel-ethanol yeast strains, an important correlation was found between *FLO11* length polymorphisms in the genome of the strains and foam production, cellular hydrophobicity, and flocculation. *FLO11* has been implicated in the interaction of yeasts with air–liquid interfaces [44,51,79,80], which is in agreement with our results that indicate the presence of yeast cells in the highly persistent foam produced by some of the industrial strains (Figures 1 and 2). Although none of the industrial strains analyzed had a *FLO11* gene with the predicted size (based on strain S288C, see Table 2), but instead had shorter version of the gene, it is also worth noting that all these strains produced biofilms (of variable sizes) that were smooth, resembling the mats and colonies produced by *flo11Δ* strains [60,65]. Not only increased central tandem repeat domains of the *FLO11* coding region were shown to be responsible for the formation of buoyant biofilms at the medium surface, but also alterations in the *FLO11* promoter region have been implicated in floating phenotypes of wild flor yeast strains [44,79,80]. Indeed, the *FLO11* promoter is one of the biggest promoters described in yeasts (~2.8 Kb), with several *cis*-regulatory elements (four activating and at least nine repressing sequences) and numerous *trans*-acting factors implicated in its

regulation, including posttranscriptional and epigenetic control by chromatin remodeling complexes, as well as prions [69,81–84]. While expression of the *FLO11* gene has been shown to be induced under limiting nutritional conditions [85,86], it will be interesting to analyze the expression of this gene during the fermenting conditions that produce high foaming phenotypes in the different genetic backgrounds of the industrial fuel-ethanol yeast strains analyzed. It is worth noting that yeast strains with flocculating phenotypes can easily appear from non-flocculating yeasts cultures [72,74,77,79,87]. Nevertheless, our results indicate that deleting the primary activator of *FLO* genes (the *FLO8* gene) from the genome of an industrial strain avoids problematic foam formation, flocculation, invasive growth, and biofilm production. From an industrial perspective, it is difficult to imagine deleting all functional flocculin genes in a selected strain to warranty single-cell budding yeasts and, thus, the strategy described in this work (*FLO8* deletion) opens new perspectives for yeast strain engineering and optimization in the growing sugarcane fuel-ethanol industry.

## 5. Conclusions

Our molecular analysis of *FLO1* and *FLO11* gene polymorphisms present in contaminant strains of *S. cerevisae* from fuel ethanol distilleries showing strong foam production during fermentation revealed significant correlations with cellular hydrophobicity, flocculation, and highly foaming phenotypes in these yeast strains. Our results also show that deleting the primary activator of *FLO* genes (the *FLO8* gene) from the genome of a contaminant and highly foaming industrial strain avoids problematic foam formation, flocculation, invasive growth, and biofilm production by the engineered strain.

**Author Contributions:** Conceptualization, H.V.d.A. and B.U.S.; investigation, methodology, and formal analysis, C.M.d.F., D.H.H., D.T. and M.d.L.B.M.; resources, H.V.d.A., M.L.L. and B.U.S.; writing—review and editing, M.L.L. and B.U.S.; project administration and funding acquisition, B.U.S. All authors have read and agreed to the published version of the manuscript.

**Funding:** This work was supported in part by grants and fellowships from the Brazilian agencies CNPq (process nos. 482905/2007-7, 143269/2009-7, 551392/2010-0, and 308389/2019-0).

**Institutional Review Board Statement:** Not applicable.

**Informed Consent Statement:** Not applicable.

**Data Availability Statement:** The data presented in this study are available in the article.

**Conflicts of Interest:** The authors declare no conflict of interest. The funders had no role in the design of the study; in the collection, analyses, or interpretation of data; in the writing of the manuscript; or in the decision to publish the results.

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
