# Peer review of "High Foam Phenotypic Diversity and Variability in Flocculant Gene Observed for Various Yeast Cell Surfaces Present as Industrial Contaminants"

_fermentation, doi:10.3390/fermentation7030127_

Round 1
Reviewer 1 Report
Title is too wordy - suggested rewording:
High foam phenotypic diversity and variability in flocculant gene observed for various yeast cell surfaces present as industrial contaminants.
A couple grammatical checks uploaded in the word document

Author Response
Reply to the Reviewer # 1 report:
We would like to thank the reviewer´s comments and suggestions, which have been all incorporated into our revised version of the manuscript, as we feel that they certainly improved the quality of our article.
Title: it was changed as suggested (High foam phenotypic diversity and variability in flocculant gene observed for various yeast cell surfaces present as industrial contaminants)
Abstract: all changes in the Abstract were done as suggested.
Grammatical checks: all changes were incorporated into the text as suggested. Since the formatting of the manuscript changed significantly, we are highlighting where the changes are located in our “fermentation-1297419-tracking changes” file.
Line 39 – improvements, including the selection of …. (now pg. 1, line 41)
Line 40 – with an increased amount …. (pg. 1, line 42)
Line 59 – and consequently, the original starter …. (pg. 2, line 18)
Line 64 – nutrient and pH, process interruptions, to mention …. (pg. 2, line 23)
Line 65 - Another highly desired characteristic … (pg. 2, line 24)
Line 66 – planktonic state to allow … (pg. 2, line 26)
Line 68 – that are a consequence of structured …. (pg. 2, line 29)
Line 284 - representative strains (CAT-1, BAT-1 ,and FT281… (pg. 7, line 24)
Line 288 – fermentation, and the maximum…. (pg. 7, line 28)
Line 289 – but soon after that, the foam …. (pg. 7, line 29)
Line 291 – but once the foam accumulated, it persisted … (pg. 7, line 30)
Line 293 – dense, and persistent foam… (pg. 7, line 32)
Line 298 - FT281, FT705, and FT714,.. (pg. 7, line 37)
Line 426 – excessive amounts of foam. Only 20% … (pg. 14, line 4)
Line 430 – The cooperation of …. (pg. 14, line 8)
Line 468 – in agreement with our results … (pg. 14, line 47)
Line 472 – and is also worth noting… (pg. 14, line 51)
Line 499 – primary activator of FLO… (pg. 15, line 14; but also see line 13 –same required change-)
Reviewer 2 Report
The manuscript by Catarina de Figueiredo, Daniella Hock and others adds to our knowledge on behaviour of yeast strains in biofuel fermentation.
It is well-written, experimentally sound and offers good documentation of results and techniques. There is also a clear separation between results and discussion. Thus, although the results do not really surprise and correlations especially between cellular hydrophobicity, flocculation and similar parameters are expected, the reviewer recommends accepting this manuscript for publication. It offers a considerably closer view on fermentation problems posed by contaminating yeast strains than before.
Author Response
Reply to the Reviewer # 2 report:
We appreciate the nice comments of the reviewer regarding our experimental approach and the results presented in our article, which addresses fermentation problems due to contaminant yeasts in the industry, a topic poorly described in the literature.